# DAC: The Double Actor-Critic Architecture for Learning Options

**Shangtong Zhang, Shimon Whiteson**
Department of Computer Science
University of Oxford
{shangtong.zhang, shimon.whiteson}@cs.ox.ac.uk

## Abstract

We reformulate the option framework as two parallel augmented MDPs. Under this novel formulation, all policy optimization algorithms can be used off the shelf to learn intra-option policies, option termination conditions, and a master policy over options. We apply an actor-critic algorithm on each augmented MDP, yielding the Double Actor-Critic (DAC) architecture. Furthermore, we show that, when state-value functions are used as critics, one critic can be expressed in terms of the other, and hence only one critic is necessary. We conduct an empirical study on challenging robot simulation tasks. In a transfer learning setting, DAC outperforms both its hierarchy-free counterpart and previous gradient-based option learning algorithms.

## 1 Introduction

Temporal abstraction (i.e., hierarchy) is a key component in reinforcement learning (RL). A good temporal abstraction usually improves exploration (Machado et al., 2017b) and enhances the interpretability of agents' behavior (Smith et al., 2018). The option framework (Sutton et al., 1999), which is commonly used to formulate temporal abstraction, gives rise to two problems: learning options (i.e., temporally extended actions) and learning a master policy (i.e., a policy over options, a.k.a. an inter-option policy).

A Markov Decision Process (MDP, Puterman 2014) with options can be interpreted as a Semi-MDP (SMDP, Puterman 2014), and a master policy is used in this SMDP for option selection. While in principle, any SMDP algorithm can be used to learn a master policy, such algorithms are data inefficient as they cannot update a master policy during option execution. To address this issue, Sutton et al. (1999) propose *intra-option* algorithms, which can update a master policy at every time step during option execution. Intra-option $Q$-Learning (Sutton et al., 1999) is a value-based intra-option algorithm and has enjoyed great success (Bacon et al., 2017; Riemer et al., 2018; Zhang et al., 2019b).

However, in the MDP setting, policy-based methods are often preferred to value-based ones because they can cope better with large action spaces and enjoy better convergence properties with function approximation. Unfortunately, theoretical study for learning a master policy with policy-based intra-option methods is limited (Daniel et al., 2016; Bacon, 2018) and its empirical success has not been witnessed. This is the first issue we address in this paper.

Recently, gradient-based option learning algorithms have enjoyed great success (Levy and Shimkin, 2011; Bacon et al., 2017; Smith et al., 2018; Riemer et al., 2018; Zhang et al., 2019b). However, most require algorithms that are customized to the option-based SMDP. Consequently, we cannot directly leverage recent advances in gradient-based policy optimization from MDPs (e.g., Schulman et al. 2015, 2017; Haarnoja et al. 2018). This is the second issue we address in this paper.

To address these issues, we reformulate the SMDP of the option framework as two augmented MDPs. Under this novel formulation, all policy optimization algorithms can be used for option learning and master policy learning off the shelf and the learning remains intra-option. We apply an actor-critic algorithm on each augmented MDP, yielding the Double Actor-Critic (DAC) architecture. Furthermore, we show that, when state-value functions are used as critics, one critic can be expressed in terms of the other, and hence only one critic is necessary. Finally, we empirically study the combination of DAC and Proximal Policy Optimization (PPO, Schulman et al. 2017) in challenging robot simulation tasks. In a transfer learning setting, DAC+PPO outperforms both its hierarchy-free counterpart, PPO, and previous gradient-based option learning algorithms.

## 2 Background

We consider an MDP consisting of a state space $\mathcal{S}$, an action space $\mathcal{A}$, a reward function $r : \mathcal{S} \times \mathcal{A} \to \mathbb{R}$, a transition kernel $p : \mathcal{S} \times \mathcal{S} \times \mathcal{A} \to [0, 1]$, an initial distribution $p_0 : \mathcal{S} \to [0, 1]$ and a discount factor $\gamma \in [0, 1)$. We refer to this MDP as $M \doteq (\mathcal{S}, \mathcal{A}, r, p, p_0, \gamma)$ and consider episodic tasks. In the option framework (Sutton et al., 1999), an option $o$ is a triple of $(\mathcal{I}_o, \pi_o, \beta_o)$, where $\mathcal{I}_o$ is an initiation set indicating where the option can be initiated, $\pi_o : \mathcal{A} \times \mathcal{S} \to [0, 1]$ is an intra-option policy, and $\beta_o : \mathcal{S} \to [0, 1]$ is a termination function. In this paper, we consider $\mathcal{I}_o \equiv \mathcal{S}$ following Bacon et al. (2017); Smith et al. (2018). We use $\mathcal{O}$ to denote the option set and assume all options are Markov. We use $\pi : \mathcal{O} \times \mathcal{S} \to [0, 1]$ to denote a master policy and consider the *call-and-return* execution model (Sutton et al., 1999). Time-indexed capital letters are random variables. At time step $t$, an agent at state $S_t$ either terminates the previous option $O_{t-1}$ w.p. $\beta_{O_{t-1}}(S_t)$ and initiates a new option $O_t$ according to $\pi(\cdot|S_t)$, or proceeds with the previous option $O_{t-1}$ w.p. $1 - \beta_{O_{t-1}}(S_t)$ and sets $O_t \doteq O_{t-1}$. Then an action $A_t$ is selected according to $\pi_{O_t}(\cdot|S_t)$. The agent gets a reward $R_{t+1}$ satisfying $\mathbb{E}[R_{t+1}] = r(S_t, A_t)$ and proceeds to a new state $S_{t+1}$ according to $p(\cdot|S_t, A_t)$. Under this execution model, we have

$$p(S_{t+1}|S_t, O_t) = \sum_a \pi_{O_t}(a|S_t)p(S_{t+1}|S_t, a),$$

$$p(O_t|S_t, O_{t-1}) = (1 - \beta_{O_{t-1}}(S_t))\mathbb{I}_{O_{t-1}=O_t} + \beta_{O_{t-1}}(S_t)\pi(O_t|S_t),$$

$$p(S_{t+1}, O_{t+1}|S_t, O_t) = p(S_{t+1}|S_t, O_t)p(O_{t+1}|S_{t+1}, O_t),$$

where $\mathbb{I}$ is the indicator function. With a slight abuse of notations, we define $r(s, o) \doteq \sum_a \pi_o(s, a)r(s, a)$. The MDP $M$ and the options $\mathcal{O}$ form an SMDP. For each state-option pair $(s, o)$ and an action $a$, we define

$$q_\pi(s, o, a) \doteq \mathbb{E}_{\pi, \mathcal{O}, p, r}[\sum_{i=1}^\infty \gamma^{i-1} R_{t+i} \mid S_t = s, O_t = o, A_t = a].$$

The state-option value of $\pi$ on the SMDP is $q_\pi(s, o) = \sum_a \pi_o(a|s)q_\pi(s, o, a)$. The state value of $\pi$ on the SMDP is $v_\pi(s) = \sum_o \pi(o|s)q_\pi(s, o)$. They are related as

$$q_\pi(s, o) = r(s, o) + \gamma \sum_{s'} p(s'|s, o)u_\pi(o, s'), \quad u_\pi(o, s') = [1 - \beta_o(s')]q_\pi(s', o) + \beta_o(s')v_\pi(s'),$$

where $u_\pi(o, s')$ is the option-value upon arrival (Sutton et al., 1999). Correspondingly, we have the optimal master policy $\pi^*$ satisfying $v_{\pi^*}(s) \geq v_\pi(s) \, \forall(s, \pi)$. We use $q_*$ to denote the state-option value function of $\pi^*$.

**Master Policy Learning:** To learn the optimal master policy $\pi^*$ given a fixed $\mathcal{O}$, one value-based approach is to learn $q_*$ first and derive $\pi^*$ from $q_*$. We can use SMDP $Q$-Learning to update an estimate $Q$ for $q_*$ as

$$Q(S_t, O_t) \leftarrow Q(S_t, O_t) + \alpha\big(\sum_{i=t}^k \gamma^{i-t} R_{i+1} + \gamma^{k-t} \max_o Q(S_k, o) - Q(S_t, O_t)\big),$$

where we assume the option $O_t$ *initiates* at time $t$ and *terminates* at time $k$ (Sutton et al., 1999). Here the option $O_t$ lasts $k-t$ steps. However, SMDP $Q$-Learning performs only one single update, yielding significant data inefficiency. This is because SMDP algorithms simply interpret the option-based SMDP as a generic SMDP, ignoring the presence of options. By contrast, Sutton et al. (1999) propose to exploit the fact that the SMDP is generated by options, yielding an update rule:

$$Q(S_t, O_t) \leftarrow Q(S_t, O_t) + \alpha[R_{t+1} + \gamma U(O_t, S_{t+1}) - Q(S_t, O_t)] \tag{1}$$

$$U(O_t, S_{t+1}) \doteq \big(1 - \beta_{O_t}(S_{t+1})\big)Q(S_{t+1}, O_t) + \beta_{O_t}(S_{t+1})\max_o Q(S_{t+1}, o).$$

This update rule is efficient in that it updates $Q$ every time step. However, it is still inefficient in that it only updates $Q$ for the executed option $O_t$. We refer to this property as *on-option*. Sutton et al. (1999) further propose Intra-option $Q$-Learning, where the update (1) is applied to *every option $o$* satisfying $\pi_o(A_t|S_t) > 0$. We refer to this property as *off-option*. Intra-option $Q$-Learning is theoretically justified only when all intra-option policies are deterministic (Sutton et al., 1999). The convergence analysis of Intra-option $Q$-Learning with stochastic intra-option policies remains an open problem (Sutton et al., 1999). The update (1) and the Intra-option $Q$-Learning can also be applied to off-policy transitions.

**The Option-Critic Architecture:** Bacon et al. (2017) propose a gradient-based option learning algorithm, the Option-Critic (OC) architecture. Assuming $\{\pi_o\}_{o \in \mathcal{O}}$ is parameterized by $\nu$ and $\{\beta_o\}_{o \in \mathcal{O}}$ is parameterized by $\phi$, Bacon et al. (2017) prove that

$$\nabla_\nu v_\pi(S_0) = \sum_{s,o} \rho(s,o|S_0,O_0) \sum_a q_\pi(s,o,a) \nabla_\nu \pi_o(a|s),$$

$$\nabla_\phi v_\pi(S_0) = -\sum_{s',o} \rho(s',o|S_1,O_0)\big(q_\pi(s',o) - v_\pi(s')\big)\nabla_\phi \beta_o(s'),$$

where $\rho$ defined in Bacon et al. (2017) is the unnormalized discounted state-option pair occupancy measure. OC is *on-option* in that given a transition $(S_t, O_t, A_t, R_{t+1}, S_{t+1})$, it updates only parameters of the executed option $O_t$. OC provides the gradient for $\{\pi_o, \beta_o\}$ and can be combined with any master policy learning algorithm. In particular, Bacon et al. (2017) combine OC with (1). Hence, in this paper, we use OC to indicate this exact combination. OC has also been extended to multi-level options (Riemer et al., 2018) and deterministic intra-option policies (Zhang et al., 2019b).

**Inferred Option Policy Gradient:** We assume $\pi$ is parameterized by $\theta$ and define $\xi \doteq \{\theta, \nu, \phi\}$. We use $\tau \doteq (S_0, A_0, S_1, A_1, \ldots, S_T)$ to denote a trajectory from $\{\pi, \mathcal{O}, M\}$, where $S_T$ is a terminal state. We use $r(\tau) \doteq \mathbb{E}[\sum_{t=1}^{T} \gamma^{t-1} R_t \mid \tau]$ to denote the total expected discounted rewards along $\tau$. Our goal is to maximize $J \doteq \int r(\tau) p(\tau) d\tau$. Smith et al. (2018) propose to interpret the options along the trajectory as *latent variables* and marginalize over them when computing $\nabla_\xi J$. In the Inferred Option Policy Gradient (IOPG), Smith et al. (2018) show

$$\nabla_\xi J = \mathbb{E}_\tau[r(\tau) \sum_{t=0}^{T} \nabla_\xi \log p(A_t|H_t)] = \mathbb{E}_\tau[r(\tau) \sum_{t=0}^{T} \nabla_\xi \log \big(\sum_o m_t(o)\pi_o(A_t|S_t)\big)],$$

where $H_t \doteq (S_0, A_0, \ldots, S_{t-1}, A_{t-1}, S_t)$ is the state-action history and $m_t(o) \doteq p(O_t = o|H_t)$ is the probability of occupying an option $o$ at time $t$. Smith et al. (2018) further show that $m_t$ can be expressed recursively via $(m_{t-1}, \{\pi_o, \beta_o\}, \pi)$, allowing efficient computation of $\nabla_\xi J$. IOPG is an *off-line* algorithm in that it has to wait for a complete trajectory before computing $\nabla_\xi J$. To admit *online* updates, Smith et al. (2018) propose to store $\nabla_\xi m_{t-1}$ at each time step and use the stored $\nabla_\xi m_{t-1}$ for computing $\nabla_\xi m_t$, yielding the Inferred Option Actor Critic (IOAC). IOAC is biased in that a stale approximation of $\nabla_\xi m_{t-1}$ is used for computing $\nabla_\xi m_t$. The longer a trajectory is, the more biased the IOAC gradient is. IOPG and IOAC are *off-option* in that given a transition $(S_t, O_t, A_t, R_{t+1}, S_{t+1})$, all options contribute to the gradient explicitly.

**Augmented Hierarchical Policy:** Levy and Shimkin (2011) propose the Augmented Hierarchical Policy (AHP) architecture. AHP reformulates the SMDP of the option framework as an augmented MDP. The new state space is $\mathcal{S}^{\text{AHP}} \doteq \mathcal{O} \times \mathcal{S}$. The new action space is $\mathcal{A}^{\text{AHP}} \doteq \mathcal{B} \times \mathcal{O} \times \mathcal{A}$, where $\mathcal{B} \doteq \{stop, continue\}$ indicates whether to terminate the previous option or not. All policy optimization algorithms can be used to learn an augmented policy

$$\pi^{\text{AHP}}\big((B_t, O_t, A_t)|(O_{t-1}, S_t)\big) \doteq \pi_{O_t}(A_t|S_t)\Big(\mathbb{I}_{B_t=cont}\mathbb{I}_{O_t=O_{t-1}} + \mathbb{I}_{B_t=stop}\pi(O_t|S_t)\Big)\Big($$

$$\mathbb{I}_{B_t=cont}(1 - \beta_{O_{t-1}}(S_t)) + \mathbb{I}_{B_t=stop}\beta_{O_{t-1}}(S_t)\Big)$$

under this new MDP, which learns $\pi$ and $\{\pi_o, \beta_o\}$ implicitly. Here $B_t \in \mathcal{B}$ is a binary random variable. In the formulation of $\pi^{\text{AHP}}$, the term $\pi(O_t|S_t)$ is gated by $\mathbb{I}_{B_t=stop}$. Consequently, the gradient for the master policy $\pi$ is non-zero only when an option terminates (also see Equation 23 in Levy and Shimkin (2011)). This suggests that the master policy learning in AHP is SMDP-style. Moreover, as suggested by the term $\pi_{O_t}(A_t|S_t)$ in $\pi^{\text{AHP}}$, the resulting gradient for an intra-option policy $\pi_o$ is non-zero only when the option $o$ is being executed (also see Equation 24 in Levy and Shimkin (2011)). This suggests that the option learning in AHP is *on-option*. Similar augmented MDP formulation is also used in Daniel et al. (2016).

# 3 Two Augmented MDPs

In this section, we reformulate the SMDP as two augmented MDPs: the high-MDP $M^{\mathcal{H}}$ and the low-MDP $M^{\mathcal{L}}$. The agent makes high-level decisions (i.e., option selection) in $M^{\mathcal{H}}$ according to $\pi, \{\beta_o\}$ and thus optimizes $\pi, \{\beta_o\}$. The agent makes low-level decisions (i.e., action selection) in $M^{\mathcal{L}}$ according to $\{\pi_o\}$ and thus optimizes $\{\pi_o\}$. Both augmented MDPs share the same samples with the SMDP $\{\mathcal{O}, M\}$.

We first define a dummy option $\#$ and $\mathcal{O}^+ \doteq \mathcal{O} \cup \{\#\}$. This dummy option is only for simplifying notations and is never executed. In the high-MDP, we interpret a state-option pair in the SMDP as a new state and an option in the SMDP as a new action. Formally speaking, we define

$$M^{\mathcal{H}} \doteq \{\mathcal{S}^{\mathcal{H}}, \mathcal{A}^{\mathcal{H}}, p^{\mathcal{H}}, p_0^{\mathcal{H}}, r^{\mathcal{H}}, \gamma\}, \quad \mathcal{S}^{\mathcal{H}} \doteq \mathcal{O}^+ \times \mathcal{S}, \quad \mathcal{A}^{\mathcal{H}} \doteq \mathcal{O},$$

$$p^{\mathcal{H}}(S_{t+1}^{\mathcal{H}}|S_t^{\mathcal{H}}, A_t^{\mathcal{H}}) \doteq p^{\mathcal{H}}\big((O_t, S_{t+1})|(O_{t-1}, S_t), A_t^{\mathcal{H}}\big) \doteq \mathbb{I}_{A_t^{\mathcal{H}}=O_t} p(S_{t+1}|S_t, O_t),$$

$$p_0^{\mathcal{H}}(S_0^{\mathcal{H}}) \doteq p_0^{\mathcal{H}}\big((O_{-1}, S_0)\big) \doteq p_0(S_0)\mathbb{I}_{O_{-1}=\#}, \quad r^{\mathcal{H}}(S_t^{\mathcal{H}}, A_t^{\mathcal{H}}) \doteq r^{\mathcal{H}}\big((O_{t-1}, S_t), O_t\big) \doteq r(S_t, O_t)$$

We define a Markov policy $\pi^{\mathcal{H}}$ on $M^{\mathcal{H}}$ as

$$\pi^{\mathcal{H}}(A_t^{\mathcal{H}}|S_t^{\mathcal{H}}) \doteq \pi^{\mathcal{H}}(O_t|(O_{t-1}, S_t)) \doteq p(O_t|S_t, O_{t-1})\mathbb{I}_{O_{t-1}\neq\#} + \pi(S_t, O_t)\mathbb{I}_{O_{t-1}=\#}$$

In the low-MDP, we interpret a state-option pair in the SMDP as a new state and leave the action space unchanged. Formally speaking, we define

$$M^{\mathcal{L}} \doteq \{\mathcal{S}^{\mathcal{L}}, \mathcal{A}^{\mathcal{L}}, p^{\mathcal{L}}, p_0^{\mathcal{L}}, r^{\mathcal{L}}, \gamma\}, \quad \mathcal{S}^{\mathcal{L}} \doteq \mathcal{S} \times \mathcal{O}, \quad \mathcal{A}^{\mathcal{L}} \doteq \mathcal{A},$$

$$p^{\mathcal{L}}(S_{t+1}^{\mathcal{L}}|S_t^{\mathcal{L}}, A_t^{\mathcal{L}}) \doteq p^{\mathcal{L}}\big((S_{t+1}, O_{t+1})|(S_t, O_t), A_t\big) \doteq p(S_{t+1}|S_t, A_t)p(O_{t+1}|S_{t+1}, O_t),$$

$$p_0^{\mathcal{L}}(S_0^{\mathcal{L}}) \doteq p^{\mathcal{L}}\big((S_0, O_0)\big) \doteq p_0(S_0)\pi(S_0, O_0), \quad r^{\mathcal{L}}(S_t^{\mathcal{L}}, A_t^{\mathcal{L}}) \doteq r^{\mathcal{L}}\big((S_t, O_t), A_t\big) \doteq r(S_t, A_t)$$

We define a Markov policy $\pi^{\mathcal{L}}$ on $M^{\mathcal{L}}$ as

$$\pi^{\mathcal{L}}(A_t^{\mathcal{L}}|S_t^{\mathcal{L}}) \doteq \pi^{\mathcal{L}}\big(A_t|(S_t, O_t)\big) \doteq \pi_{O_t}(A_t|S_t)$$

We consider trajectories with nonzero probabilities and define $\Omega \doteq \{\tau \mid p(\tau|\pi, \mathcal{O}, M) > 0\}$, $\Omega^{\mathcal{H}} \doteq \{\tau^{\mathcal{H}} \mid p(\tau^{\mathcal{H}}|\pi^{\mathcal{H}}, M^{\mathcal{H}}) > 0\}$, $\Omega^{\mathcal{L}} \doteq \{\tau^{\mathcal{L}} \mid p(\tau^{\mathcal{L}}|\pi^{\mathcal{L}}, M^{\mathcal{L}}) > 0\}$. With $\tau \doteq (S_0, O_0, S_1, O_1, \dots, S_T)$, we define a function $f^{\mathcal{H}} : \Omega \to \Omega^{\mathcal{H}}$, which maps $\tau$ to $\tau^{\mathcal{H}} \doteq (S_0^{\mathcal{H}}, A_0^{\mathcal{H}}, S_1^{\mathcal{H}}, A_1^{\mathcal{H}}, \dots, S_T^{\mathcal{H}})$, where $S_t^{\mathcal{H}} \doteq (O_{t-1}, S_t), A_t^{\mathcal{H}} \doteq O_t, O_{-1} \doteq \#$. We have:

**Lemma 1** $p(\tau|\pi, \mathcal{O}, M) = p(\tau^{\mathcal{H}}|\pi^{\mathcal{H}}, M^{\mathcal{H}})$, $r(\tau) = r(\tau^{\mathcal{H}})$, and $f^{\mathcal{H}}$ is a bijection.

*Proof.* See supplementary materials.

We now take action into consideration. With $\tau \doteq (S_0, O_0, A_0, S_1, O_1, A_1 \dots, S_T)$, we define a function $f^{\mathcal{L}} : \Omega \to \Omega^{\mathcal{L}}$, which maps $\tau$ to $\tau^{\mathcal{L}} \doteq (S_0^{\mathcal{L}}, A_0^{\mathcal{L}}, S_1^{\mathcal{L}}, A_1^{\mathcal{L}}, \dots, S_T^{\mathcal{L}})$, where $S_t^{\mathcal{L}} \doteq (S_t, O_t), A_t^{\mathcal{L}} \doteq A_t$. We have:

**Lemma 2** $p(\tau|\pi, \mathcal{O}, M) = p(\tau^{\mathcal{L}}|\pi^{\mathcal{L}}, M^{\mathcal{L}})$, $r(\tau) = r(\tau^{\mathcal{L}})$, and $f^{\mathcal{L}}$ is a bijection.

*Proof.* See supplementary materials.

**Proposition 1**

$$J \doteq \int r(\tau)p(\tau|\pi, \mathcal{O}, M)d\tau = \int r(\tau^{\mathcal{H}})p(\tau^{\mathcal{H}}|\pi^{\mathcal{H}}, M^{\mathcal{H}})d\tau^{\mathcal{H}} = \int r(\tau^{\mathcal{L}})p(\tau^{\mathcal{L}}|\pi^{\mathcal{L}}, M^{\mathcal{L}})d\tau^{\mathcal{L}}.$$

*Proof.* Follows directly from Lemma 1 and Lemma 2.

Lemma 1 and Lemma 2 indicate that sampling from $\{\pi, \mathcal{O}, M\}$ is equivalent to sampling from $\{\pi^{\mathcal{H}}, M^{\mathcal{H}}\}$ and $\{\pi^{\mathcal{L}}, M^{\mathcal{L}}\}$. Proposition 1 indicates that optimizing $\pi, \mathcal{O}$ in $M$ is equivalent to optimizing $\pi^{\mathcal{H}}$ in $M^{\mathcal{H}}$ and optimizing $\pi^{\mathcal{L}}$ in $M^{\mathcal{L}}$. We now make two observations:

**Observation 1** $M^{\mathcal{H}}$ *depends on* $\{\pi_o\}$ *while* $\pi^{\mathcal{H}}$ *depends on* $\pi$ *and* $\{\beta_o\}$.

**Observation 2** $M^{\mathcal{L}}$ *depends on* $\pi, \{\beta_o\}$ *while* $\pi^{\mathcal{L}}$ *depends on* $\{\pi_o\}$.

Observation 1 suggests that when we keep the intra-option policies $\{\pi_o\}$ fixed and optimize $\pi^{\mathcal{H}}$, we are implicitly optimizing $\pi$ and $\{\beta_o\}$ (i.e., $\theta$ and $\phi$). Observation 2 suggests that when we keep the master policy $\pi$ and the termination conditions $\{\beta_o\}$ fixed and optimize $\pi^{\mathcal{L}}$, we are implicitly optimizing $\{\pi_o\}$ (i.e., $\nu$). All policy optimization algorithms for MDPs can be used off the shelf to optimize the two actors $\pi^{\mathcal{H}}$ and $\pi^{\mathcal{L}}$ with samples from $\{\pi, \mathcal{O}, M\}$, yielding a new family of algorithms for master policy learning and option learning, which we refer to as the Double Actor-Critic (DAC) architecture. Theoretically, we should optimize $\pi^{\mathcal{H}}$ and $\pi^{\mathcal{L}}$ alternatively with different samples to make sure $M^{\mathcal{H}}$ and $M^{\mathcal{L}}$ are stationary. In practice, optimizing $\pi^{\mathcal{H}}$ and $\pi^{\mathcal{L}}$ with the same samples improves data efficiency. The pseudocode of DAC is provided in the supplementary materials. We present a thorough comparison of DAC, OC, IOPG and AHP in Table 1. DAC combines the advantages of both AHP (i.e., compatibility) and OC (intra-option learning). Enabling off-option learning of intra-option policies in DAC as IOPG is a possibility for future work.

|      | Learning $\pi$ | Learning $\{\pi_o, \beta_o\}$ | Online Learning | Compatibility |
|------|----------------|-------------------------------|-----------------|---------------|
| AHP  | SMDP           | on-option                     | yes             | yes           |
| OC   | intra-option   | on-option                     | yes             | no            |
| IOPG | intra-option   | off-option                    | no              | no            |
| DAC  | intra-option   | on-option                     | yes             | yes           |

Table 1: A comparison of AHP, OC, IOPG and DAC. (1) For learning $\{\pi_o, \beta_o\}$, all four are intra-option. (2) IOAC is online with bias introduced and consumes extra memory. (3) Compatibility indicates whether a framework can be combined with any policy optimization algorithm off-the-shelf.

In general, we need two critics in DAC, which can be learned via all policy evaluation algorithms. However, when state value functions are used as critics, Proposition 2 shows that the state value function in the high-MDP ($v_{\pi^{\mathcal{H}}}$) can be expressed by the state value function in the low-MDP ($v_{\pi^{\mathcal{L}}}$), and hence only one critic is needed.

**Proposition 2** $v_{\pi^{\mathcal{H}}}\big((o, s')\big) = \sum_{o'} \pi^{\mathcal{H}}\big(o'|(o, s')\big) v_{\pi^{\mathcal{L}}}\big((s', o')\big)$, where

$$v_{\pi^{\mathcal{H}}}\big((o, s')\big) \doteq \mathbb{E}_{\pi^{\mathcal{H}}, M^{\mathcal{H}}}\big[\sum_{i=1}^{\infty} \gamma^{i-1} R_{t+i}^{\mathcal{H}} \mid S_t^{\mathcal{H}} = (o, s')\big],$$
$$v_{\pi^{\mathcal{L}}}\big((s', o')\big) \doteq \mathbb{E}_{\pi^{\mathcal{L}}, M^{\mathcal{L}}}\big[\sum_{i=1}^{\infty} \gamma^{i-1} R_{t+i}^{\mathcal{L}} \mid S_t^{\mathcal{L}} = (s', o')\big],$$

*Proof.* See supplementary materials.

Our two augmented MDP formulation differs from the one augmented MDP formulation in AHP mainly in that we do not need to introduce the binary variable $B_t$. It is this elimination of $B_t$ that leads to the intra-option master policy learning in DAC and yields a useful observation: the call-and-return execution model (with Markov options) is similar to the polling execution model (Dietterich, 2000), where an agent reselects an option *every time step* according to $\pi^{\mathcal{H}}$. This observation opens the possibility for more intra-option master policy learning algorithms. Note the introduction of $B_t$ is necessary if one would want to formulate the option SMDP as a single augmented MDP and apply standard control methods from the MDP setting. Otherwise, the augmented MDP will have $\pi^{\mathcal{H}}$ in both the augmented policy and the new transition kernel. By contrast, in a canonical MDP setting, a policy does not overlap with the transition kernel.

**Beyond Intra-option $Q$-Learning:** In terms of learning $\pi$ with a fixed $\mathcal{O}$, Observation 1 suggests we optimize $\pi^{\mathcal{H}}$ on $M^{\mathcal{H}}$. This immediately yields a family of policy-based algorithms for learning a master policy, all of which are intra-option. Particularly, when we use Off-Policy Expected Policy Gradients (Off-EPG, Ciosek and Whiteson 2017) for optimizing $\pi^{\mathcal{H}}$, we get all the merits of both Intra-option $Q$-Learning and policy gradients for free. (1) By definition of $M^{\mathcal{H}}$ and $\pi^{\mathcal{H}}$, Off-EPG optimizes $\pi$ in an intra-option manner and is as data efficient as Intra-option $Q$-Learning. (2) Off-EPG is an off-policy algorithm, so off-policy transitions can also be used, as in Intra-option $Q$-Learning. (3) Off-EPG is off-option in that all the options, not only the executed one, explicitly contribute to the policy gradient at every time step. Particularly, this off-option approach does not require deterministic intra-option policies like Intra-option $Q$-Learning. (4) Off-EPG uses a policy for decision making, which is more robust than value-based decision making. We leave an empirical study of this particular application for future work and focus in this paper on the more general problem, learning $\pi$ and $\mathcal{O}$ simultaneously. When $\mathcal{O}$ is not fixed, the MDP ($M^{\mathcal{H}}$) for learning $\pi$ becomes non-stationary. We, therefore, prefer on-policy methods to off-policy methods.

# 4 Experimental Results

We design experiments to answer the following questions: (1) Can DAC outperform existing gradient-based option learning algorithms (e.g., AHP, OC, IOPG)? (2) Can options learned in DAC translate into a performance boost over its hierarchy-free counterparts? (3) What options does DAC learn?

DAC can be combined with any policy optimization algorithm, e.g., policy gradient (Sutton et al., 2000), Natural Actor Critic (NAC, Peters and Schaal 2008), PPO, Soft Actor Critic (Haarnoja et al., 2018), Generalized Off-Policy Actor Critic (Zhang et al., 2019a). In this paper, we focus on the combination of DAC and PPO, given the great empirical success of PPO (OpenAI, 2018). Our PPO implementation uses the same architecture and hyperparameters reported by Schulman et al. (2017).

Levy and Shimkin (2011) combine AHP with NAC and present an empirical study on an inverted pendulum domain. In our experiments, we also combine AHP with PPO for a fair comparison. To the best of our knowledge, this is the first time that AHP has been evaluated with state-of-the-art policy optimization algorithms in prevailing deep RL benchmarks. We also implemented IOPG and OC as baselines. Previously, Klissarov et al. (2017) also combines OC and PPO in PPOC. PPOC updates $\{\pi_o\}$ with a PPO loss and updates $\{\beta_o\}$ in the same manner as OC. PPOC applies vanilla policy gradients directly to train $\pi$ in an intra-option manner, which is not theoretically justified. We use 4 options for all algorithms, following Smith et al. (2018). We report the online training episode return, smoothed by a sliding window of size 20. All curves are averaged over 10 independent runs and shaded regions indicate standard errors. All implementations are made publicly available [1]. More details about the experiments are provided in the supplementary materials.

## 4.1 Single Task Learning

We consider four robot simulation tasks used by Smith et al. (2018) from OpenAI gym (Brockman et al., 2016). We also include the combination of DAC and A2C (Clemente et al., 2017) for reference. The results are reported in Figure 1.

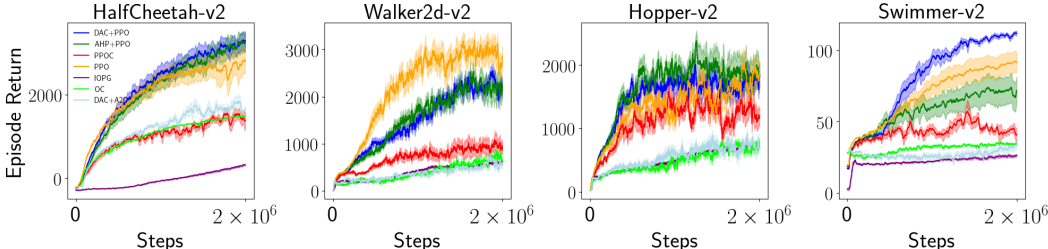

Figure 1: Online performance on a single task

**Results:** (1) Our implementations of OC and IOPG reach similar performance to that reported by Smith et al. (2018), which is significantly outperformed by both vanilla PPO and option-based PPO (i.e., DAC+PPO, AHP+PPO). However, the performance of DAC+A2C is similar to OC and IOPG. These results indicate that the performance boost of DAC+PPO and AHP+PPO mainly comes from the more advanced policy optimization algorithm (PPO). This is exactly the major advantage of DAC and AHP. They allow all state-of-the-art policy optimization algorithms to be used off the shelf to learn options. (2) The performance of DAC+PPO is similar to vanilla PPO in 3 out of 4 tasks. DAC+PPO outperforms PPO in Swimmer by a large margin. This performance similarity between an option-based algorithm and a hierarchy-free algorithm is expected and is also reported by Harb et al. (2018); Smith et al. (2018); Klissarov et al. (2017). Within a single task, it is usually hard to translate the automatically discovered options into a performance boost, as primitive actions are enough to express the optimal policy and learning the additional structure, the options, may be overhead. (3) The performance of DAC+PPO is similar to AHP+PPO, as expected. The main advantage of DAC over AHP is its data efficiency in learning the master policy. Within a single task, it is possible that an agent focuses on a "mighty" option and ignores other specialized options, making master policy learning less important. By contrast, when we switch tasks, cooperation among different options

becomes more important. We, therefore, expect that the data efficiency in learning the master policy in DAC translates into a performance boost over AHP in a transfer learning setting.

## 4.2 Transfer Learning

We consider a transfer learning setting where after the first 1M training steps, we switch to a new task and train the agent for other 1M steps. The agent is not aware of the task switch. The two tasks are correlated and we expect learned options from the first task can be used to accelerate learning of the second task.

We use 6 pairs of tasks from DeepMind Control Suite (DMControl, Tassa et al. 2018): `CartPole` = (`balance`, `balance_sparse`), `Reacher` = (`easy`, `hard`), `Cheetah` = (`run`, `backward`), `Fish` = (`upright`, `downleft`), `Walker1` = (`squat`, `stand`), `Walker2` = (`walk`, `backward`). Most of them are provided by DMControl and some of them we constructed similarly as Hafner et al. (2018). The maximum score is always 1000. More details are provided in the supplementary materials. There are other possible paired tasks in DMControl but we found that in such pairs, PPO hardly learns anything in the second task. Hence, we omit those pairs from our experiments. The results are reported in Figure 2.

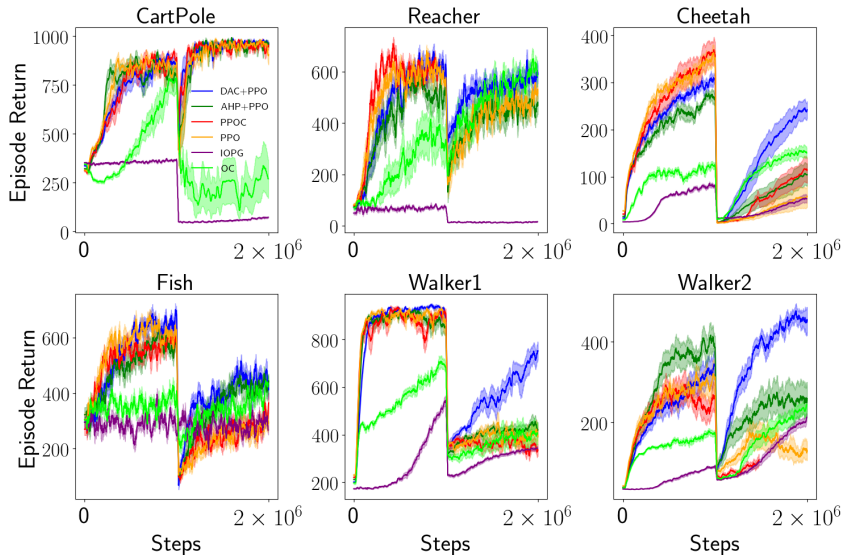

Figure 2: Online performance for transfer learning

**Results:** (1) During the first task, DAC+PPO consistently outperforms OC and IOPG by a large margin and maintains a similar performance to PPO, PPOC, and AHP+PPO. These results are consistent with our previous observations in the single task learning setting. (2) After the task switch, the advantage of DAC+PPO becomes clear. DAC+PPO outperforms all other baselines by a large margin in 3 out of 6 tasks and is among the best algorithms in the other 3 tasks. This satisfies our previous expectation about DAC and AHP in Section 4.1. (3) We further study the influence of the number of options in `Walker2`. Results are provided in the supplementary materials. We find 8 options are slightly better than 4 options and 2 options are worse. We conjecture that 2 options are not enough for transferring the knowledge from the first task to the second.

## 4.3 Option Structures

We visualize the learned options and option occupancy of DAC+PPO on `Cheetah` in Figure 3. There are 4 options in total, displayed via different colors. The upper strip shows the option occupancy during an episode at the end of the training of the first task (`run`). The lower strip shows the option occupancy during an episode at the end of the training of the second task (`backward`). Both episodes last 1000 steps.[2] The four options are distinct. The blue option is mainly used when the cheetah is

"flying". The green option is mainly used when the cheetah pushes its left leg to move right. The yellow option is mainly used when the cheetah pushes its left leg to move left. The red option is mainly used when the cheetah pushes its right leg to move left. During the first task, the red option is rarely used. The cheetah uses the green and yellow options for pushing its left leg and uses the blue option for flying. The right leg rarely touches the ground during the first episode. After the task switch, the flying option (blue) transfers to the second task, the yellow option specializes for moving left, and the red option is developed for pushing the right leg to the left.

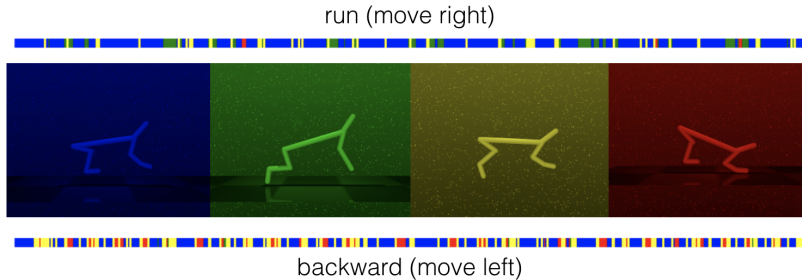

Figure 3: Learned options and option occupancy of DAC+PPO in `Cheetah`

## 5   Related Work

Many components in DAC are not new. The idea of an augmented MDP is suggested by Levy and Shimkin (2011); Daniel et al. (2016). The augmented state spaces $S^{\mathcal{H}}$ and $S^{\mathcal{L}}$ are also used by Bacon et al. (2017) to simplify the derivation. Applying vanilla policy gradient to $\pi^{\mathcal{L}}$ and $M^{\mathcal{L}}$ leads immediately to the Intra-Option Policy Gradient Theorem (Bacon et al., 2017). The augmented policy $\pi^{\mathcal{H}}$ is also used by Smith et al. (2018) to simplify the derivation and is discussed in Bacon (2018) under the name of mixture distribution. Bacon (2018) discusses two mechanisms for sampling from the mixture distribution: a two-step sampling method (sampling $B_t$ first then $O_t$) and a one-step sampling method (sampling $O_t$ directly), where the latter can be viewed as an expected version of the former. The two-step one is implemented by the call-and-return model and is explicitly modelled by Levy and Shimkin (2011) via introducing $B_t$, which is not used in either Bacon (2018) or our work. Bacon (2018) mentions that the one-step modelling can lead to reduced variance compared to the two-step one. However, there is another significant difference: the one-step modelling is more data efficient than the two-step one. The two-step one (e.g., AHP) yields SMDP learning while the one-step one (e.g., our approach) yields intra-option learning (for learning the master policy). This difference is not recognized in Bacon (2018) and we are the first to establish it, both conceptually and experimentally. Although the underlying chain of DAC is the same as that of Levy and Shimkin (2011); Daniel et al. (2016); Bacon et al. (2017); Bacon (2018), DAC is the first to formulate the two augmented MDPs explicitly. It is this explicit formulation that allows the off-the-shelf application of all state-of-the-art policy optimizations algorithm and combines advantages from both OC and AHP.

The gradient of the master policy first appeared in Levy and Shimkin (2011). However, due to the introduction of $B_t$, that gradient is nonzero only if an option terminates. It is, therefore, SMDP-learning. The gradient of the master policy later appeared in Daniel et al. (2016) in the probabilistic inference method for learning options, which, however, assumes a linear structure and is off-line learning. The gradient of the master policy also appeared in Bacon (2018), which is mixed with all other gradients. Unless we work on the augmented MDP directly, we cannot easily drop in other policy optimization techniques for learning the master policy, which is our main contribution and is not done by Bacon (2018). Furthermore, that policy gradient is never used in Bacon (2018). All the empirical study uses Q-Learning for the master policy. By contrast, our explicit formulation of the two augmented MDPs generates a family of online policy-based intra-option algorithms for master policy learning, which are compatible with general function approximation.

Besides gradient-based option learning, there are also other option learning approaches based on finding bottleneck states or subgoals (Stolle and Precup, 2002; McGovern and Barto, 2001; Silver and Ciosek, 2012; Niekum and Barto, 2011; Machado et al., 2017a). In general, these approaches are expensive in terms of both samples and computation (Precup, 2018).

Besides the option framework, there are also other frameworks to describe hierarchies in RL. Dietterich (2000) decomposes the value function in the original MDP into value functions in smaller MDPs in the MAXQ framework. Dayan and Hinton (1993) employ multiple managers on different levels for describing a hierarchy. Vezhnevets et al. (2017) further extend this idea to FeUdal Networks, where a manager module sets abstract goals for workers. This goal-based hierarchy description is also explored by Schmidhuber and Wahnsiedler (1993); Levy et al. (2017); Nachum et al. (2018). Moreover, Florensa et al. (2017) use stochastic neural networks for hierarchical RL. We leave a comparison between the option framework and other hierarchical RL frameworks for future work.

# 6 Conclusions

In this paper, we reformulate the SMDP of the option framework as two augmented MDPs, allowing in an off-the-shelf application of all policy optimization algorithms in option learning and master policy learning in an intra-option manner.

In DAC, there is no clear boundary between option termination functions and the master policy. They are different internal parts of the augmented policy $\pi^{\mathcal{H}}$. We observe that the termination probability of the active option becomes high as training progresses, although $\pi^{\mathcal{H}}$ still selects the same option. This is also observed by Bacon et al. (2017). To encourage long options, Harb et al. (2018) propose a cost model for option switching. Including this cost model in DAC is a possibility for future work.

**Acknowledgments**

SZ is generously funded by the Engineering and Physical Sciences Research Council (EPSRC). This project has received funding from the European Research Council under the European Union's Horizon 2020 research and innovation programme (grant agreement number 637713). The experiments were made possible by a generous equipment grant from NVIDIA. The authors thank Matthew Smith and Gregory Farquhar for insightful discussions. The authors also thank anonymous reviewers for their valuable feedbacks.

## Footnotes

[1] https://github.com/ShangtongZhang/DeepRL

[2] The video of the two episodes is available at `https://youtu.be/KOZP-HQtx6M`

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
