[Supplementary Material]

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

# A    Assumptions and Proofs

## A.1    Assumptions

We use standard assumptions (Sutton et al., 1999; Bacon et al., 2017) about the MDP and options. Particularly, we assume all options are Markov.

## A.2    Proof of Lemma 1

*Proof.*

$$
\begin{aligned}
p(\tau|\pi, \mathcal{O}, M) &= p(S_0)\Pi_{t=0}^{T-1}p(O_t|S_t, O_{t-1})p(S_{t+1}|S_t, O_t) \\
&= p(S_0)\Pi_{t=0}^{T-1}p(O_t|S_t, O_{t-1})\mathbb{I}_{A_t^{\mathcal{H}}=O_t}p(S_{t+1}|S_t, O_t) \quad \text{(Definition of } A_t^{\mathcal{H}}) \\
&= p^{\mathcal{H}}(S_0^{\mathcal{H}})\Pi_{t=0}^{T-1}\pi^{\mathcal{H}}(A_t^{\mathcal{H}}|S_t^{\mathcal{H}})p^{\mathcal{H}}(S_{t+1}^{\mathcal{H}}|S_t^{\mathcal{H}}, A_t^{\mathcal{H}}) \quad \text{(Definition of } \pi^{\mathcal{H}} \text{ and } p^{\mathcal{H}}) \\
&= p(\tau^{\mathcal{H}}|\pi^{\mathcal{H}}, M^{\mathcal{H}})
\end{aligned}
$$

$r(\tau) = r(\tau^{\mathcal{H}})$ follows directly from the definition of $r^{\mathcal{H}}$. $f^{\mathcal{H}}$ is an injection by definition. The definition of $p^{\mathcal{H}}$ guarantees $f^{\mathcal{H}}$ is a surjection. So $f^{\mathcal{H}}$ is a bijection.

## A.3    Proof of Lemma 2

*Proof.*

$$
\begin{aligned}
p(\tau|\pi, \mathcal{O}, M) &= p(S_0)p(O_0|S_0)\Pi_{t=0}^{T-1}p(A_t|S_t, O_t)p(S_{t+1}|S_t, A_t)p(O_{t+1}|S_{t+1}, O_t) \\
&= p^{\mathcal{L}}(S_0^{\mathcal{L}})\Pi_{t=0}^{T-1}\pi^{\mathcal{L}}(A_t^{\mathcal{L}}|S_t^{\mathcal{L}})p^{\mathcal{L}}(S_{t+1}^{\mathcal{L}}|S_t^{\mathcal{L}}, A_t^{\mathcal{L}}) \quad \text{(Definition of } \pi^{\mathcal{L}} \text{ and } p^{\mathcal{L}}) \\
&= p(\tau^{\mathcal{L}}|\pi^{\mathcal{L}}, M^{\mathcal{L}})
\end{aligned}
$$

$r(\tau) = r(\tau^{\mathcal{L}})$ follows directly from the definition of $r^{\mathcal{L}}$. $f^{\mathcal{L}}$ is an injection by definition. The definition of $p^{\mathcal{L}}$ guarantees $f^{\mathcal{L}}$ is a surjection. So $f^{\mathcal{L}}$ is a bijection.

## A.4    Proof of Proposition 2

*Proof.*

$$
\begin{aligned}
v_{\pi^{\mathcal{H}}}\big((o, s')\big) &= \mathbb{E}_{\pi^{\mathcal{H}}, M^{\mathcal{H}}}\Big[\sum_{i=1}^{\infty}\gamma^{i-1}R_{t+i}^{\mathcal{H}} \mid S_t^{\mathcal{H}} = (o, s')\Big] \\
&= \mathbb{E}_{\pi, \mathcal{O}, M}\Big[\sum_{i=1}^{\infty}\gamma^{i-1}R_{t+i} \mid O_{t-1} = o, S_t = s'\Big] \\
&= u_\pi(o, s') \quad \text{(Definition of } u_\pi \text{ in Sutton et al. (1999))} \\
&= [1 - \beta_o(s')]q_\pi(s, o) + \beta_o(s')\sum_{o'}q_\pi(s', o') \\
&= \sum_{o'}\pi^{\mathcal{H}}\big(o'|(o, s')\big)q_\pi(s', o') \\
&= \sum_{o'}\pi^{\mathcal{H}}\big(o'|(o, s')\big)\mathbb{E}_{\pi, \mathcal{O}, M}\Big[\sum_{i=1}^{\infty}\gamma^{i-1}R_{t+i} \mid S_t = s', O_t = o'\Big] \\
&= \sum_{o'}\pi^{\mathcal{H}}\big(o'|(o, s')\big)\mathbb{E}_{\pi^{\mathcal{L}}, M^{\mathcal{L}}}\Big[\sum_{i=1}^{\infty}\gamma^{i-1}R_{t+i}^{\mathcal{L}} \mid S_t^{\mathcal{L}} = (s', o')\Big] \\
&= \sum_{o'}\pi^{\mathcal{H}}\big(o'|(o, s')\big)v_{\pi^{\mathcal{L}}}\big((s', o')\big)
\end{aligned}
$$

# B  Details of Experiments

## B.1  Pseudocode of DAC

Pseudocode of DAC is provided in Algorithm 1.

---

**Algorithm 1:** Pseudocode of DAC

---

**Input:**
Parameterized $\pi, \{\pi_o, \beta_o\}_{o \in \mathcal{O}}$
Policy optimization algorithms $\mathbb{A}_1, \mathbb{A}_2$

Get an initial state $S_0$
$t \leftarrow 0$
**while** *True* **do**
    Sample $O_t$ from $\pi^{\mathcal{H}}\big(\,\cdot\,|(O_{t-1}, S_t)\big)$
    Sample $A_t$ from $\pi^{\mathcal{L}}\big(\,\cdot\,|(S_t, O_t)\big)$
    Execute $A_t$, get $R_{t+1}, S_{t+1}$
    `// The two optimizations can be done in any order or alternatively`
    Optimize $\pi^{\mathcal{H}}$ with $(S_t^{\mathcal{H}}, A_t^{\mathcal{H}}, R_{t+1}, S_{t+1}^{\mathcal{H}})$ and $\mathbb{A}_1$
    Optimize $\pi^{\mathcal{L}}$ with $(S_t^{\mathcal{L}}, A_t^{\mathcal{L}}, R_{t+1}, S_{t+1}^{\mathcal{L}})$ and $\mathbb{A}_2$
    $t \leftarrow t + 1$
**end**

---

## B.2  Details of Environments

`CartPole` consists of `balance` and `balance_sparse`, where the latter has a sparse reward. `Reacher` consists of `easy` and `hard`, where the latter has a smaller target sphere than the former. Those four tasks are provided in DMControl. `Cheetah` consists of `run` and `backward`. The former is from DMControl, the latter is from Hafner et al. (2018), where the horizontal speed of the cheetah is negated before being used for computing rewards. In this task, the cheetah is encouraged to run backward rather than forward. `Fish` consists of `upright` and `downleft`. The former is from DMControl. In the latter, we negate the uprightness before using it to compute rewards. This task encourages the fish to be "downleft". `Walker1` consists of `squat` and `stand`. The latter is from DMControl, where a reward is given when the torso height of the walker is larger than $1.2$. In the former, we give a reward when the torso height is larger than $0.6$. `Walker2` consists of `walk` and `backward`. The former is from DMControl. In the latter, we negate the horizontal speed as in `Cheetah-backward`.

## B.3  Implementation Details

Open AI Gym and DMControl are available at `https://gym.openai.com/` and `https://github.com/deepmind/dm_control`.

**Function Parameterization:** We base our parameterization on Schulman et al. (2017). For an option $o$, $\beta_o$ is parameterized as a two-hidden-layer network. A sigmoid activation function is used after the output layer. $\pi_o$ is parameterized as a two-hidden-layer network. A Tanh activation function is used after the output layer to output the mean of the Gaussian policy $\pi_o$. The std of $\pi_o$ is a state-independent variable as Schulman et al. (2015, 2017). The master policy $\pi$ is parameterized in the same manner as $\pi_o$ except that a softmax function is used after the output layer. The value function $q_\pi$ has the same parameterization as $\pi$ except that the activation function after the output layer is linear. All hidden layers have 64 hidden units. For Mujoco tasks, we use a Tanh activation for hidden layers as suggested by Schulman et al. (2017). For DMControl tasks, we find a ReLU (Nair and Hinton, 2010) activation for hidden layers produces better performance. We use this parameterization for all compared algorithms.

**Preprocessing:** States are normalized by a running estimation of mean and std.

**Hyperparameter Tuning:** Our PPO implementation is based on the PPO implementation from Dhariwal et al. (2017). Our DAC+PPO and AHP+PPO implementations inherited common hyper-parameters from the PPO implementation. The DAC+A2C implementation is based on the A2C implementation from Dhariwal et al. (2017). The OC and IOPG implementations are based on Smith et al. (2018) and hyperparameters of A2C from Dhariwal et al. (2017). All the implementations have 4 options following Smith et al. (2018).

**Hyperparameters of PPO:**
Optimizer: Adam with $\epsilon = 10^{-5}$ and an initial learning rate $3 \times 10^{-4}$
Discount ratio $\gamma$: 0.99
GAE coefficient: 0.95
Gradient clip by norm: 0.5
Rollout length: 2048 environment steps
Optimization epochs: 10
Optimization batch size: 64
Action probability ratio clip: 0.2

**Additional Hyperparameters of DAC+PPO and AHP+PPO:**
When optimizing the high MDP, we apply an entropy regularizer with weight 0.01. When optimizing the low MDP, we do not use an entropy regularizer. For DAC+PPO, we perform optimizations for the two MDPs with the same data. For each MDP, we perform 5 optimization epochs. So the overall amount of gradient updates remains the same as PPO. Our preliminary experiments show that performing the two optimizations alternatively leads to a similar performance.

**Additional Hyperparameters of PPOC:**
Entropy regularizer wieght: 0.01
Option switching penalty weight: 0.01, as suggested by Harb et al. (2018)

**Hyperparameters of DAC+A2C:**
Number of workers: 4, as used by Smith et al. (2018)
Optimizer: Adam with $\epsilon = 10^{-5}$ and an initial learning rate $3 \times 10^{-4}$
Discount ratio $\gamma$: 0.99
GAE coefficient: 0.95
Gradient clip by norm: 0.5
Entropy regularizer wieght: 0.01
Rollout length: 5 environment steps

**Hyperparameters of IOPG:**
Number of workers: 4, as used by Smith et al. (2018)
Optimizer: Adam with $\epsilon = 10^{-5}$ and an initial learning rate $3 \times 10^{-4}$
Discount ratio $\gamma$: 0.99
Gradient clip by norm: 0.5

**Hyperparameters of OC:**
Number of workers: 4, as used by Smith et al. (2018)
Optimizer: Adam with $\epsilon = 10^{-5}$ and an initial learning rate $3 \times 10^{-4}$
Discount ratio $\gamma$: 0.99
Gradient clip by norm: 0.5
Rollout length: 5 environment steps
Entropy regularizer wieght: 0.01
Probability for selecting a random option: 0.1, as used by Harb et al. (2018)
Target network update frequency: $10^3$ optimization steps
Option switching penalty weight: 0.01, as suggested by Harb et al. (2018)

**Computing Infrastructure:**
We conducted our experiments on an Nvidia DGX-1 with PyTorch.

## B.4  Other Experimental Results

Figure 4 studies the influence of the number of options on performance. In the first task, the performance is similar. In the second task, 8 options are slightly better than 4 options, while 2 options are clearly worse.

Figure 4: The influence of number of options on performance.