[Reviews · NeurIPS 2019]

Reviewer 1



Post-rebuttal update: I have read the rebuttal. Thanks for the clarification regarding they type of experiments where there is a larger gap between DAC and the baselines, as well as the clarification on PPO+OC/IOPG. The paper proposes a new method for learning options in a hierarchical reinforcement learning set-up. The method works by decomposing the original problem into two MDPs, that can each be solved using conventional policy-based methods. This allows new state-of-the-art methods to easily be 'dropped in' to improve HRL. Originality. The paper clearly builds off of earlier results like the option-critic framework. Nevertheless, the paper proposes new original ideas that are contribute to this line of work and that admit further extension. Quality: The technical quality is high. I did not find inaccuracies in the main material, although I didn't read the supplementary material in detail. The experiments provide appropriate support for the claimed benefits of the proposed method. The experiments seem reproducible and reliable (based on multiple random seed, with standard errors provided for all results). From the experimental results, it would be interesting to analyze on what kind of tasks DAC clearly performs best, and on what types of tasks it is comparable to other methods like AHP+ PPO. I was surprised to see the relative performance of OC and IOPG looking quite different compared to the reported results in Smith et al, but that doesn't have a bearing on the conclusions of the paper. It would also be interesting to see how easy or hard it is to modify other approaches (OC or IOPG) to use a method like PPO rather than vanilla gradients. Clarity. The paper is in general well-written and well-structured. Equations and experimental results are presented in a clear and interpretable fashion. Some minor comments are provided below. Significance: I think the proposed method can significantly impact HRL research. The ability to drop in state of the art optimizers is very interesting and make it easy to further improve results in the future. Minor comments: l. 27 "there are no policy-based intra-option algorithms for learning a master policy". These seems to contradict the overview in table 1. Perhaps this statement could be made more specific. l. 45. The definitions seem to be specific to discrete MDPs (eg specification of the co-domain of p as [0,1]). However, later on continuous MDPs are considered as well. l. 62-63. In the equation for u_pi, on the RHS, I think the second s should be s'? l 155-156. Maybe the statement here can be made more practical. I don't know any practical policy optimization algorithm that can be guaranteed go improve the policy, due the stochastic nature of the gradients. E.g. in case gradients of the base algorithm are unbiased, is there anything we can say about the combined pi^H and pi^L updates? l 173-174. Polling can also be seen as a special case of call-and-return (where B is always equal to 'stop'). Maybe this statement can be made more clear.

Reviewer 2



"Equation 24 in Levy and Shimkin (2011))." --> it's better if the paper is self-contained. Fig.2 --> why in most of the cases DAC is not performing well on the first task? "4.3 Option Structure" --> I am not entirely sure what do we learn from this section as it's rather descriptive and there is no conclusion to be made (and no baseline to compare with, or no way to quantify the performance) Fig. 1 --> Why DAC outperforms the baselines only on two tasks out of four? Rows 227-228 "The main advantage of DAC over AHP is its data efficiency in learning the master policy." --> it's not clear how this statement is supported by the experimental results. Rows 249-250 "DAC+PPO outperforms all other baselines by a large margin in 3 out of 6 tasks and is among the best algorithms in the other 3 tasks." --> that does not support the statement provided in the abstract. Also, would be nice to define what is a large margin. Rows 253-254 "We conjecture that 2 options are not enough for transferring the knowledge from the first task to the second." --> what can we learn from it?

Reviewer 3



I believe that the result presented in this paper follows from Levy and Shimkin (2011) and Bacon (2017), up to a change of notation (and terminology). It suffices to see that the call-and-return model with Markov options leads to a chain where the probability of choosing the next option conditioned on the augmented state is: $P(O_t|S_t, O_{t-1}) = (1 - \beta_{O_{t-1}}(S_t))1_{O_{t-1}=O_t} + \beta_{O_{t-1}}(S_t)\mu(O_t|S_t)$. Note that $P(O_t|S_t, O_{t-1})$ can be seen as a policy over options; one which also happens to contain the termination functions. The form of this chain is the same one behind the intra-option methods, presented in the value-based context, but independent of any algorithm. Notation: Writing $\pi_{O_t}(S_t, \cdot)$ suggests that you have a joint distribution over state and actions while pi really is a conditional distribution over options. $\pi_{O_t}(\cdot | S_t)$ is less ambiguous. > Unfortunately, to the best of our knowledge, there are no policy-based intra-option algorithms for learning a master policy. See Daniel et al. (2016), Bacon's thesis (2018) where the gradient for the policy over options is shown, and Levy and Shimkin (2011) which boil down to the same chain structure (albeit different notation). [Update:] I maintain the same overall score. The paper could eventually help other researcher to gain more clarity in this topic. It would however need to be properly positioned in the lineage of Levy and Shimkin (2011), Daniel et al. (2016) and Bacon et al. (2017), which all share the same underlying Markov chain. The "double actor-critic" point of view belongs in that same family. All these papers are also based on the "intra-option" point of view: the one that you get when assuming Markov options.

Reviewer 4



Post rebuttal update: I have read the rebuttal prepared by the authors and believe that they have made a good job in clarifying some of the points that I brought up in my original review. I also believe that the authors have the opportunity to significantly improve their paper by incorporating some of the comments made in their rebuttal; e.g., to clarify some the points that were not clear in their original submission and some of the statements which were, perhaps, too strong, in particular regarding the empirical performance/advantages of the proposed method. ------ This paper introduces a method to learn option policies and their termination conditions by re-expressing the corresponding resulting SMDP as two MDPs defined over augmented state and action spaces. The authors introduce a mathematical formulation involving two MDPs that model the different challenges of learning hierarchical policies: a high-level MDP that models the problem of learning a policy over options and their corresponding termination conditions; and a low-level MDP that models the problem of learning an individual policy for each option. The authors also show that the value function of one MDP can be expressed in terms of the value function of the other, so that a single critic is required. This is an interesting paper based on a variation of an existing method/formulation called Augmented Hierarchical Policy (AHP). It is relatively well written and the experiments demonstrate (1) that using the proposed method either improves performance (when used to solve a single task) or that it doesn't hurt performance; and (2) that, when evaluated on transfer learning settings, the learned options are often reusable across tasks. I have few comments and questions: 1) I suggest toning down the sentences in the abstract and introduction that describe the empirical performance/advantages of the method (e.g. when the authors say, in line 8, that "DAC outperforms previous gradient-based option learning learning algorithms by a large margin and significantly outperforms its hierarchy-free counterparts in a transfer learning setting"). This claim is inconsistent with, e.g. line 220 ("The performance of DAC+PPO is similar to vanilla PPO in 3 out of 4 tasks"), line 247 ("[DAC] maintains a similar performance to PPO, PPOC, and AHP+PPO"), and line 250 ("[DAC works best] in 3 out of 6 tasks"). 2) in the Background section, you introduce two different probabilities: p(s_{t+1}, o_{t+1} | s_t, o_t) and p(o_t, s_{t+1} | o_{t-1}, s_t). Why is the 2nd definition needed, if (assuming that I understand this correctly) only the 1st one is used to express the equations that follow those definitions? Where is the latter probability used/required? 3) to make notation more consistent, I suggest making the dependence of q(s,o,a) on the policy pi explicit, just like you did when defining q_pi(s,o) and v_pi(s). 4) in the equations immediately after line 84, the variables S0, O0 and S1 were used but were never defined/introduced. What are they? 5) it is not immediately clear to me why, in the AHP formulation, the policy is defined over an action space given by (B x O x A). In particular, if the policy pi^{AHP} already specifies what option o_t should be selected at time t, why does it need to also specify the primitive action a_t that should be executed? Shouldn't the primitive action a_t be picked stochastically by the policy of the selected option? 6) the introduction of an option "#" that is never executed (line 124) is not well-motivated. Why is it necessary to include it in the action space of the high-level MDP? And what is O_{-1}? 7) in Table 1, what do you mean by the "Compatibility" column? In particular, what does it mean for an algorithm to be compatible? 8) I am not sure I understand the decision of using an on-policy method (vs an off-policy method) "[given that] O is not fixed [and that therefore] the MDP M^H for learning pi becomes non-stationary" (lines 187-189). What is the relation between an MDP being non-stationary and an on-policy method being more appropriate? 9) I am curious about the possible relation between (a) the observation that when using DAC the termination probability often becomes higher as training progresses (line 296); and (b) the option occupancies shown in Fig3. In particular, why is it that during the first task, a single option is selected most of the time (the blue one), while during the execution of the second task, options alternate very fast even during the execution of each "part" of the Cheetah movement? Is this related to the fact that the termination probabilities are already very close to 1 when the second task begins?

[Author Response · NeurIPS 2019]

First we would like to thank all the reviewers for their time and comments.

**Reviewer 1: (a)** We conjecture DAC will outperform AHP when it is important to coordinate among different options.
This could be the case when the task is complicated. In our experiments, Cheetah and Walker (1&2), where the
advantage of DAC+PPO is clear, are usually considered more complicated than CartPole and Reacher. **(b)** PPOC tries to
combine OC with PPO and is compared in our paper, although that combination is not theoretically sound. Combining
IOPG and PPO is even harder. As discussed by Smith et al., this combination requires products of importance sampling
ratios to correct trajectory distributions, which could yield high variance. **(c)** Thanks for pointing out other minor issues.
This is indeed very constructive and we shall fix typos and make statements clearer accordingly in the final version.

**Reviewer 2: (Related Work)** We would like to clarify that our work builds on the option framework and we compared
to all related works (e.g., OC, IOPG, PPOC). Particularly, we also compared with AHP, which is 8 years old and
was never evaluated in deep RL setting. As other HRL frameworks usually do not have option termination functions,
their ability to express a hierarchy is different from the option framework. They thus are not directly comparable.
**(Performance)** In the single task setting and the first phase of the transfer learning setting, DAC is similar to other
baselines. This phenomenon is expected. Usually we do not expect an option-based method to outperform option-free
counterparts in a single task, as learning options adds overhead unless useful options can be learned easily. This
performance similarity is also observed in previous option-based works (e.g., OC, IOPG, PPOC). In the second phase
of the transfer setting, DAC outperforms all baselines in 50% tasks and is no worse than any other baseline in the other
50% tasks. We consider this to be a solid achievement but we'll tone down our wording and make it less ambiguous in
the final version. **(Visualization)** We visualize option structures following the convention of OC and IOPG. All those
visualizations are descriptive and aim to understand the behavior of the agent. **(AHP)** The reviewer pointed out that
the data efficiency of DAC over AHP is not directly supported by experimental results. We will clarify that this data
efficiency is the major difference between DAC and AHP and comes from the formalization directly (intra-option vs.
SMDP). If the performance of DAC and AHP are different, this should be the major cause. We shall examine other
possible minor implementation differences but they are unlikely to contribute to the performance difference.

**Reviewer 3: (Master Policy)** We agree that our $\pi^H$ is similar to Bacon's mixture distribution (Sec 3.5, Bacon 2018),
which is also used in IOPG. Bacon discussed two mechanisms for sampling from the mixture distribution: a two-step
sampling method and a one-step sampling method. The latter can be viewed as an expected version of the former. The
two-step one is implemented by the call-and-return model and is explicitly modelled in AHP (Levy and Shimkin, 2011)
via introducing an extra variable. This new variable is not used in either Bacon's thesis or our work. Bacon's thesis
mentions that the one-step modelling can lead to reduced variance compared to the two-step one. However, there is
another significant difference: the one-step modelling is more data efficient than the two-step one. The two-step one
(e.g., AHP) yields SMDP learning, where the policy gradient of the master policy is non-zero only when an option
terminates. (This is a side effect of the extra variable in AHP.) The one-step one (e.g., our approach) yields intra-option
learning, where the policy gradient is non-zero every step. This difference is not recognized in Bacon's thesis and
we are the first to establish it, both conceptually and experimentally. **(Policy Gradients)** We agree that the gradient
of the master policy appears in Bacon's thesis, which, however, is mixed with all other gradients. Unless we work
on the augmented MDP directly, we cannot easily drop in other policy optimization techniques, which is our second
contribution and is not done in Bacon's thesis. Furthermore, that policy gradient is never used in Bacon's thesis. All the
experiments use Q-Learning for the master policy. We are happy to rephrase our claim about the policy gradient and
more explicitly acknowledge Bacon's thesis. Doing so does not reduce our main contribution, a framework where 1) we
can drop in all advanced policy optimization techniques 2) we can maintain intra-option learning for both the master
policy and options. Bacon's thesis achieves 2) not 1); AHP achieves 1) not 2). We have achieved both. Although our
underlying chain is the same as Bacon's thesis after algebraic manipulation, our new formulation brings in new insights
for combining the option framework and other advanced policy optimization techniques in a data efficient way.

**Reviewer 4: (1)** We will reword our claim to make it less ambiguous, as in our rebuttal Reviewer 2(Performance). **(2)**
It is not used. We shall remove it. **(3)** We agree. We shall change it accordingly. **(4)** They are defined in L52-56, e.g.,
$S_1$ is the state at step 1. **(5)** AHP tries to model the call-and-return sampling strategy explicitly in one single augmented
MDP, which is the reason that the triple $(O, B, A)$ is used as a new action. **(6)** At the starting state $S_0$, we do not have
a previous option (the initial option $O_0$ is the current option). However, $\pi^H$ depends on both the current state and
the previous option (if it exists). To make a consistent expression for $\pi^H$, we introduced this placeholder # and $O_{-1}$.
**(7)** As indicated in the caption of Table 1, we say an algorithm is compatible if it is possible to drop in other policy
optimization techniques (instead of vanilla policy gradient) directly. **(8)** If an MDP is non-stationary, old transitions
may belong to a different transition kernel. Off-policy methods, however, usually require a fixed transition kernel. **(9)**
There is a tendency for an agent to use a single option within one task as learning other options may involve overhead.
This could be the reason why termination goes closely to one at the end of the first task. When the task changes, the
previously used single option is likely to be not so useful. So the agent has the motivation to build different new options
and use them. It can achieve this by selecting the same option every step.

[Meta-Review · NeurIPS 2019]

The paper introduces a double actor critic architecture for learning options. The authors define 2 augmented MDPs for learning the option selection policy as well as the options themselves. Using this MDP formulation, off-the-shelf policy learning algorithms can be used for learning option selection as well as option policies, which was not possible with previous algorithms. Both hierachy levels where optimized with PPO. The reviews for this paper are borderline. Most reviewers appreciated the intutive idea and the promising results reported in the paper. The biggest concern raised by R3 was in terms of novelty of the approach as similar augmented markov chains have been already used in Levy and Shimkin (2011), Daniel et al. (2016) and Bacon et al. (2017). However, after reading the paper I agree with the authors that modelling options as 2 MDPs is not explicitely done in these approaches (disallowing to use off-the-shelf policy learning algorithms for both levels) or an extra variable (termination event) is introduced, which affects the sample efficiency of the algorithm. While these relations have been clarified in the rebuttal, it is not clearly stated in the paper. However, I trust the authors can properly positioning their work within the existing literature for the final version and recommend acceptance as I found the idea intuitive and the experiments convincing.